# Tumor-Suppressive Functions of the Aryl Hydrocarbon Receptor (AhR) and AhR as a Therapeutic Target in Cancer

**DOI:** 10.3390/biology12040526

**Published:** 2023-03-30

**Authors:** Daniel J. Elson, Siva K. Kolluri

**Affiliations:** 1Cancer Research Laboratory, Department of Environmental and Molecular Toxicology, Oregon State University, Corvallis, OR 97331, USA; 2Linus Pauling Institute, Oregon State University, Corvallis, OR 97331, USA

**Keywords:** aryl hydrocarbon receptor, endogenous, differentiation, co-regulators, tumor suppression, xenobiotics, pluripotency, genetically engineered mouse models, xenografts

## Abstract

**Simple Summary:**

Cancer is driven by the excessive activity of growth-promoting genes and deficient activity of the genes that restrain cell growth. Genes that suppress cell growth possess tumor-suppressive activity. Knowing how these tumor-suppressive genes function and pharmacological methods to restore or elicit their growth-inhibiting activity is of keen interest for therapeutic development. The aryl hydrocarbon receptor (AhR) is a ligand-activated transcription factor, and the activation of AhR by different molecules drives a wide range of biological effects, with both adverse and beneficial outcomes. Certain molecules that bind to AhR elicit tumor-suppressive effects (e.g., selective growth inhibition of cancer cells, apoptosis). Loss of AhR expression leads to increased tumorigenesis in different mouse models. Therapeutic targeting of the receptor requires insights into the molecular mechanisms that lead to tumor suppression, the determinants of the response to AhR ligands, and the cancer types that are responsive to AhR-selective modulators.

**Abstract:**

The aryl hydrocarbon receptor (AhR) is a ligand-activated transcription factor involved in regulating a wide range of biological responses. A diverse array of xenobiotics and endogenous small molecules bind to the receptor and drive unique phenotypic responses. Due in part to its role in mediating toxic responses to environmental pollutants, AhR activation has not been traditionally viewed as a viable therapeutic approach. Nonetheless, the expression and activation of AhR can inhibit the proliferation, migration, and survival of cancer cells, and many clinically approved drugs transcriptionally activate AhR. Identification of novel select modulators of AhR-regulated transcription that promote tumor suppression is an active area of investigation. The development of AhR-targeted anticancer agents requires a thorough understanding of the molecular mechanisms driving tumor suppression. Here, we summarized the tumor-suppressive mechanisms regulated by AhR with an emphasis on the endogenous functions of the receptor in opposing carcinogenesis. In multiple different cancer models, the deletion of AhR promotes increased tumorigenesis, but a precise understanding of the molecular cues and the genetic targets of AhR involved in this process is lacking. The intent of this review was to synthesize the evidence supporting AhR-dependent tumor suppression and distill insights for development of AhR-targeted cancer therapeutics.

## 1. Introduction

The aryl hydrocarbon receptor (AhR) is a ligand-activated transcription factor involved in a wide range of developmental, physiological, and disease processes, including cancer [1,2,3,4,5,6,7,8]. While extensive work has defined the negative impacts associated with the activation of AhR by ligands such as polycyclic aromatic hydrocarbons (PAHs) and dioxins, AhR activation can also promote desirable biological endpoints such as tumor suppression and immunomodulation [8,9,10,11]. The purpose of this article was to summarize and discuss the tumor-suppressive functions of AhR. While emphasis is placed on synthesizing mechanistic insights related to the endogenous–rather than xenobiotic–effects mediated by the receptor, the ligand-dependent functions in tumor suppression are also discussed. Further, we summarized and discussed the regulatory mechanisms controlling AhR’s transcriptional activity to guide molecular studies targeting the receptor.

There are significantly different and ostensibly conflicting reports of AhR functioning as either a pro-tumorigenic or a tumor-suppressive factor in cancer, and this has been reviewed [12,13]. The intent of this review was to specifically address the biological and molecular contexts where AhR acts as a tumor suppressor in order to provide a framework for targeting AhR as a therapeutic strategy. A lucid understanding of the molecular pathways regulated by AhR in the context of tumor suppression is essential for targeting the receptor and defining the molecular vulnerabilities that engender sensitivity to the activation of AhR.

## 2. Review of AhR Biology and Signaling

### 2.1. Role of AhR in Xenobiotic Metabolism

AhR belongs to the superfamily of transcription factors containing both basic helix–loop–helix (bHLH) and PER-ARNT-SIM (PAS) domains (14). The PAS domain is conserved across all kingdoms of life, and functions in sensing and activating biological responses to environmental changes. Related transcription factors containing the PAS domain, such as hypoxia-inducible factor 1α (HIF1α), and circadian clock proteins such as PER1 and BMAL, respond to environmental cues such as oxygen tension and light [14]. In this respect, the aryl hydrocarbon receptor functions as both an internal and external sensor of chemical signals.

The role of AhR in the cellular detoxification or metabolism of small molecules is well established [15]. Exposure to xenobiotics, or activation by endogenous metabolites, induces the receptor’s activation and a subsequent transcriptional response that then attempts to bring the cell back to a state of homeostasis. A major component of this homeostatic response is induction of the cytochrome P450 enzymes, including CYP1A1, CYP1A2, and CYP1B1, which function, through substrate oxidation, to convert the ligands to more water-soluble forms that can be conjugated by Phase II enzymes (e.g., glutathione-S-transferase) and excreted via Phase III enzymatic transport [16]. While it was evolutionarily acquired as a transcriptional circuit to ameliorate xenobiotic insults, exposure to certain xenobiotics can induce AhR-mediated toxicity. It is in this context that AhR has been historically and extensively studied, as AhR was originally cloned and identified as the receptor responsible for mediating the biological effects of dioxin (2,3,7,8-tetrachloro-dibenzo-*p*-dioxin (TCDD)) [17], a legacy anthropogenic pollutant. TCDD binds with high-affinity to AhR, is resistant to CYP1-mediated metabolic degradation, and is highly lipophilic. Together, these properties make TCDD highly bio-accumulative, and its toxic effects are from chronic AhR activation [18].

Within an organism, activation and antagonism of AhR have diverse consequences that depend critically on the cellular abundance and characteristics of the initiating ligand, as well as specific tissue and organ dynamics. While AhR is a master regulator of xenobiotic metabolism, it also has important roles in processes such as development, differentiation, and immune function.

### 2.2. The AhR Signaling Pathway

The transcriptionally inactive AhR complex is located in the cytosol, sequestered by an HSP90 dimer [19,20], co-chaperone p23 [21,22], AhR-interacting protein (AIP), and the protein kinase Src [23]. The complex functions to both sequester AhR in the cytosol and hold the AhR in conformation to interact with ligands [24]. Upon an agonistic ligand binding to the receptor, AhR-associated AIP dissociates from the receptor, exposing its nuclear localization sequence, and importin-β then binds to AhR and transports the receptor into the nucleus.

Once in the nucleus, AhR can then interact with the heterodimer partner, aryl hydrocarbon receptor nuclear translocator (Arnt), also known as HIF-1β, where the AhR–Arnt complex then binds to the regulatory regions of DNA containing the consensus motif (5′-TNGCGTG-3′) [25,26] or the xenobiotic response element (XRE) (also known as dioxin response element (DRE), or the AhR response element (AhRE) and modulates the transcription of the target gene. While XRE-driven transcription through AhR–Arnt signaling represents the most well studied mode of regulation, AhR can modulate gene expression through non-XRE elements [27,28] and can also interact with other transcription factors including c- MAF [29], KLF6 [30], RelA [31], and other NF-κB complex members [32].

Structurally, AhR contains the following features, proceeding from its amino terminus: a bHLH domain (amino acids (a.a.) 33–87); PAS-A (a.a. 111–273) and PAS-B domains (a.a. 275–386); and a transactivation domain (TAD) that contains an acidic region, a glutamine-rich (Q-rich) region, and a proline/serine-rich region (a.a. 490–805) [33,34]. AhR binds to the regulatory regions of DNA via its bHLH domain [33], which is masked by one molecule of HSP90 in the unliganded cytoplasmic complex [19], partially overlapping with the PAS-A region. The second molecule of HSP90 interacts dominantly with the PAS-B region. The bHLH domain dimerizes with the bHLH of Arnt to form a bundle of four helixes that interact with the XRE [25]. Similarly, the PAS-A domains of AhR and Arnt function as a dimerization interface, conferring specificity and enhancing the complex’s stability and DNA binding [35,36]. The PAS-B domain functions as the ligand-binding domain [36] and is required for nuclear translocation of the receptor, but is dispensable for heterodimerization and ligand-binding [37,38]. Notably, high-affinity ligand-binding and the subsequent induction of metabolizing enzymes represents a vertebrate adaptation and has been suggested as an acquired function in response to aromatic marine natural products [1]. Early invertebrate AhR paralogs in *Caenhorabiditis elegans* [39,40] and *Drosophila melanogaster* [41] function primarily in neuronal and sensory development, and this role is retained in vertebrates in addition to its ligand-mediated activities [42].

Species-specific [43], ligand-specific [44,45], dose-dependent [46], tissue-specific, and microenvironment-specific factors [47,48] together influence the apical biological outcome downstream of the activation of the AhR ligand, so making initial predictions of a ligand’s effects in a given model system is extremely difficult. For example, species-specific divergences in the receptor’s structure and ligand-binding affinity result in significant differences in the transcriptional [43] and phenotypic response to ligands such as TCDD between humans [43], mice [49], and guinea pigs [50]. The mouse AhR differs from its human ortholog in its ligand-binding affinity due to a single amino acid substitution in the ligand-binding pocket [51]. Consequently, TCDD binds with significantly greater affinity (approximately 10×) to the mouse AhR, while distinct ligands such as indoxyl-3-sulfate (of endogenous origin) bind with greater affinity to the human receptor [51,52]. To understand some of these interspecies differences, our laboratory previously generated an AhR homology model for the AhR PAS domain [53,54].

### 2.3. Role of Co-Regulators in AhR Signaling

The presence or absence of co-regulatory proteins and their activity levels influence both the basal transcription and ligand inducibility of AhR [55,56] (Figure 1). Recently, hexokinase 2 was found to be a transcriptional target of AhR and a positive regulator of AhR-mediated transcription [57]. Important regulators of AhR signaling also include proteins such as AhRR [58], TiPARP [59,60,61,62], Gadd45b [63], p300, CREB binding protein (p300/CBP) [64], Smad3 [65], and SIN3A [66,67]. AhRR and TiPARP function to provide negative feedback on AhR’s transcriptional activity, while SIN3A, typically regarded as a transcriptional repressor, was found to function as an essential co-activator for the transcription of CYP1A1 [67]. An additional mechanism fine-tuning the function of AhR was recently identified [68]. Bourner et al. found that differential expression levels of ARNT isoforms altered the transcriptional activity of AhR, and ARNT isoform-dependent regulation of AhR was shown to be dependent on phosphorylation by the AhR target gene casein kinase 2 [68].

We recently discovered that the expression of the cell cycle inhibitor and direct AhR target gene, p27^Kip1^, represses AhR-mediated transcription [69] (Figure 2). Loss of p27^Kip1^ in lung cancer cells resulted in significantly enhanced basal and ligand-induced levels of AhR target genes such as CYP1A1 and AHRR, and this negative regulation was found for the transcription of p27^Kip1^ itself [69]. Interestingly, p27^Kip1^-dependent transcriptional repression was previously found to operate via the formation of a protein complex containing mouse retinoblastoma protein (Rb) homolog p130, mSIN3A, E2F4, and multiple histone deacetylases (HDACs 1, 4, and 5) [70,71], Further, AhR’s interaction with Rb [72] and transcriptional regulation of AhR downstream from Rb have been reported [73], indicating bidirectional regulation between AhR and components of the cell cycle’s machinery.

The p300/CBP enzymes function as transcriptional co-activators via their acetyltransferase activity, and the catalytic activity of p300/CBP was found to be essential for AhR’s transcriptional activity [64]. Notably, ARNT was identified as the key target of p300/CBP’s acetylation for activating transcription [64]. Interestingly, previous reports indicated that the acetylation of the p27^Kip1^ by p300/CBP-associated factor (PCAF) attenuated its transcriptional repressor activity and promoted its degradation, suggesting an additional layer of regulatory control [74,75].

## 3. AhR-Driven Tumor Suppression by Cancer Type

Cancer is fundamentally driven by the hyperactivation of growth factor signaling pathways (oncogenes), coupled with the loss of negative feedback from regulators that would attenuate the proliferative signals (tumor suppressor genes). In this respect, AhR, as a transcription factor that controls a wide array of genes, is capable of both inhibiting oncogene expression [76] and activating the expression of tumor suppressor genes [77,78,79]. The evolutionarily conserved function of AhR in directing development and differentiation is mirrored in cancer cells, where the expression and activation of AhR can negatively regulate the proliferation of stem cell and control cell fate [80,81,82].

In different cancer types and genetic models, AhR has been shown to negatively regulate signaling pathways and factors which promote tumorigenesis (Figure 3). Examples include Wnt/β-catenin signaling [83,84], TGFβ and SHH signaling [85,86], pluripotency factors [87,88,89], epigenetic regulators such as BET proteins [90], and estrogen receptor α (ERα) [91,92]. AhR can also regulate the progression of the cell cycle through interactions with checkpoint regulators such as Rb [72,93] and activating the expression of cell cycle inhibitors such as cyclin G2 [77] and the cyclin-dependent kinase inhibitors p27^Kip1^ [69,78,94] and p21^Cip1^ [79].

Studies of AhR knockout mice from multiple laboratories found that AhR-null are mice viable but display multiple developmental defects such as hepatic portal fibrosis [95,96], impaired fertility [97,98], altered immune function [95,99], and decreased liver and body size during the first four weeks of life [100]. While exhibiting altered development, AhR-null mice do not spontaneously develop tumors. This indicates that the endogenous tumor-suppressive activities of AhR become apparent in the context of oncogene activation and disabled function of the tumor suppressor genes. Below, we review and discuss the tissue- and cancer-specific contexts where AhR has been shown to suppress carcinogenesis and inhibit tumor growth. An emphasis has been placed on cancer types where there are in vivo studies supporting AhR-dependent tumor suppression.

### 3.1. Prostate Cancer

Fritz et al. used the TRAMP mouse model (transgenic adenocarcinoma of mouse prostate) to investigate differences in tumorigenesis in *AHR*^+/+^, *AHR*^+/−^, and *AHR*^−/−^ backgrounds [101]. The TRAMP mouse model is driven by prostate-specific expression of the simian virus 40 (SV40) large T, and small t antigens, which disable the tumor-suppressive activity of p53 and Rb. *AHR*^−/−^ or AhR heterozygous mice had significantly greater tumor incidence relative to their AhR wild-type counterparts in TRAMP backgrounds. The ability of AhR to sequester ARNT, as the obligate heterodimer partner of the pro-angiogenic factor HIF-1α, has been proposed as a tumor-suppressive mechanism conferred by the expression of AhR. In support of this phenomenon, studies in TRAMP mice found that the expression of AhR opposed vanadate-induced production of vascular endothelial growth factor (VEGF) [102]. In mice, treatment with the selective AhR modulator 6-methyl-1,3,8-trichlorobenzofuran (6-MCDF) inhibited the metastasis of prostate tumors and decreased the production of VEGF [103]. In LnCaP xenografts, activation of AhR by the tryptophan metabolite 2-(1′H-indole-3′-carbonyl)-thiazole-4-carboxylic acid methyl ester (ITE) was found to potently suppress tumor growth [88].

### 3.2. Lung Cancer

Lung fibroblasts isolated from *AHR*^−/−^ mice exhibited increased phosphorylation of Akt, indicating that the expression of AhR may oppose growth factor signaling downstream of PI3K-Akt [104]. Consistent with this effect, in a genetic mouse model of non-small cell lung cancer (NSCLC) driven by a glycine to aspartate mutation (G to D) in the *K-Ras* oncogene (*K-Ras*^G12D)^, *AHR*^−/−^ mice developed more tumors than their *AHR*^+/+^ counterparts and featured increased stem cell populations characterized by the expression of pluripotency markers such as *MYCC*, *SOX2*, and *NANOG*, together with the increased expression of progenitor cell markers [105].

A role of AhR in suppressing lung cancer cell metastasis and migration has also been elucidated [106,107,108]. Tsai et al. found that the expression of AhR functioned to inhibit autophagy and the migration of NSCLC cells, and that the inhibition of autophagy was driven by AhR’s interaction with and degradation of BNIP through the ubiquitin–proteasome system [108]. Nothdurft et al. identified AhR as a regulatory factor opposing metastasis through an unbiased genome-wide knockdown approach, where mice implanted with AhR knockdown tumor cells exhibited significantly increased metastases and poorer survival [106]. Further, knockdown of AhR expression induced the epithelial-to-mesenchymal transition and increased the invasiveness via enhanced TGF-β signaling. Notably, the authors discovered that the activation of AhR by omeprazole drove the suppression of lung cancer cells’ growth via an AhR-dependent induction of the expression of activating transcription factor 4 (ATF4) and upregulation of asparagine synthetase (ASNS) [106]. Previous studies have reported the AhR-dependent induction of ER stress and the activation of ATF4 [109].

We identified benzimidazoisoquinolines (BBQs) as a class of high-affinity, rapidly metabolized AhR ligands that do not exhibit in vivo toxicity and drive AhR-dependent immunomodulation [8,110,111,112]. In NSCLC, we recently discovered 11-chloro-7H-benzimidazo[2,1-a]benzo[de]iso-quinolin-7-one (11-Cl-BBQ) as a ligand that exhibited AhR-dependent antiproliferative effects [94]. The activation of AhR by 11-Cl-BBQ induced the expression of the cell cycle inhibitors p21^Cip1^, p27^Kip1^, and CABLES1 (a newly identified cyclin-dependent kinase inhibitor) [113,114] and activated p53 signaling [94]. Importantly AhR, p27^Kip1^, and p53 were found to be required for the G1 phase cell cycle arrest downstream of 11-Cl-BBQ [94].

### 3.3. Intestinal Cancers

Multiple studies have supported a tumor-suppressive role for the expression of AhR in cancers of the intestine [83,84,115,116,117,118,119,120], and AhR has emerged as a key regulator of intestinal homeostasis [121,122,123]. Broadly, AhR appears to oppose carcinogenesis in the colon via downregulation of the inflammatory responses [10,118,119,121,124,125] inhibiting stem cell proliferation via the repression of factors such as FOXM1 and Wnt/β-catenin [84,115,126], and, by virtue of controlling these processes, promoting proper differentiation during regeneration after colonic injury [115,122,126].

Different laboratories have investigated the impact of deficiencies in AhR in intestinal cancers using colon cancer models that are driven by colonic inflammation and tumor promotion downstream of chemical insults, pathogen infections [10,118,125], or high-fat diets [116]. For instance, in a mouse model of colitis-associated tumors, colonic epithelial cell-specific deletion of the expression of AhR in the colon resulted in increased stem cell proliferation in the intestinal crypts, increased organoid-forming efficiency when colonocytes were cultured ex vivo, and led to a greater incidence and size of adenomas and adenocarcinomas [115]. The expression of AhR promotes the integrity of the colonic barrier, and dietary AhR ligands such as indole-3-carbinol were found to be protective against inflammation and malignant transformation [125].

The anticarcinogenic effects of AhR have also been highlighted in the *APC^Min/+^* colon cancer mouse model, driven by the loss of the tumor-suppressive function of adematous polyposis coli (APC), resulting in excess stem cell proliferation downstream from the increased β-catenin levels. Multiple studies have found that the loss of AhR in APC mutant genetic backgrounds drove increased intestinal tumorigenesis [83,84]. Han et al. found that *APC^S580/+^* and *K-Ras^G12D/+^* mutations drove increased intestinal tumorigenesis and reduced survival in mice with a colonic epithelial cell-specific deletion of AhR [84]. The ability of AhR to restrict the proliferation of colonic stem cells was also observed by Metidiji et al., where AhR-dependent transcriptional induction of the E3 ubiquitin ligases Rnf43, and Znrf3 negatively regulated Wnt/β–catenin signaling [125].

### 3.4. Medulloblastoma/Neuroblastoma/Glioblastoma

In cancers of the nervous system, the activity of AhR has been shown to inhibit sonic hedgehog (SHH)-driven medulloblastoma through blocking TGF-β signaling. Deletion of the SHH-repressor Patched (*PTCH1*) in mice results in lethal medulloblastomas [127], and the deletion of *AHR* in *PTCH1^−/−^* granule cerebellar progenitor cells led to arrested differentiation, exacerbated tumorigenesis, and significantly reduced survival (median survival: 63.5 days for AHR^+/+^ versus 33 days for AHR^−^^/−^) [86]. In neuroblastoma, AhR has been shown to inhibit tumors’ growth and metastasis [128]. The expression of AhR is inversely correlated with the expression of the neuroblastoma driver *MYCN* [87], and the overexpression or activation of AhR by the endogenous metabolite kynurenine inhibited the growth of xenografts [129]. Interestingly, the endogenous steroids 3α,5α-tetrahydrocorticosterone and 3α,5β-tetrahydrocorticosterone (5α- and 5β-THB) were recently identified as physiologically relevant AhR ligands with roles in promoting neuronal development and differentiation, and these compounds were also found to induce the differentiation of neuroblastoma cells [129]. In patient-derived glioblastoma cells, the deletion of AhR resulted in enhanced xenograft growth, invasion, and the expression of migratory genes [130]. Furthermore, AhR was found to be a direct repressor of the expression of Oct-4 in glioblastoma cancer stem cells, where treatment with the tryptophan derivative ITE promoted the differentiation of GBM CSC and suppressed the growth of GBM xenografts [88].

### 3.5. Liver Cancer

Multiple studies have indicated a role of the expression of AhR in suppressing liver carcinogenesis [131,132,133], while chronic activation of AhR by metabolically recalcitrant ligands such as TCDD promoted carcinogenesis in rodent models, and this has been extensively studied [12]. *AHR^−/−^* and *AHR^+/−^* mice were found to be more sensitive to chemically induced hepatocarcinogenesis caused by the mutagen diethylnitrosamine (DEN), relative to their wild-type counterparts [133]. Similarly, the expression of AhR restricted liver carcinogenesis following toxic injury with CCl_4_, and *AHR^-/-^* mice featured increased stem populations with greater expression of Wnt pathway mediators, Axin2, Dkk1, and cyclin D1, and greater expression and nuclear localization of β-catenin [131]. Consistent with the observations of Fan et al., *AHR^-/-^* mice treated with DEN exhibited a significantly greater tumor burden relative to their *AHR^+/+^* counterparts [131]. In HepG2 xenografts, the activation of AhR by the tryptophan metabolite ITE was found to potently suppress tumor growth [88].

### 3.6. Leukemia

There are indications that AhR can function as a tumor suppressor in blood cancers, primarily through regulating the differentiation status of monocytic or lymphoid progenitors. The identification of stemregenin-1 (SR-1), as an AhR antagonist that promotes the expansion of hematopoietic stem cells (HSCs), indicated that AhR has an essential role in initiating the differentiation of HSCs [81]. Antagonizing AhR with SR-1 resulted in the expansion of promyelocytic leukemia cells [134]. In acute myeloid leukemia (AML), AhR functions to promote differentiation [90,135], and AhR was found to a positive regulator of the response to BETi therapy (inhibitors of bromodomain and extraterminal proteins) in AML [90]. Additional evidence in HL-60 cells has supported the pro-differentiation role of AhR in leukemia [89,136,137].

### 3.7. Melanoma

In melanoma cells, the knockdown of AhR’s expression resulted in increased expression of the stem cell marker ALDH1a1, and the increased ALDH1a1 was found to drive enhanced invasion, migration, and tumorigenecity [138]. Knockdown of AhR and ALDH1a1 reduced these hallmarks [138]. Similarly, the expression of AhR was found to inhibit primary tumorigenesis, migration, and invasion in B16F10 melanoma cells when injected into AhR^+/+^ but not AhR*^−^*^/*−*^ mice [139]. Melanoma cells expressing a constitutively active AhR were found to have reduced tumorigenicity in either genetic background. We previously found that the anti-inflammatory drug leflunomide had AhR-dependent antiproliferative effects in melanoma cells [140,141].

### 3.8. Breast Cancer

In breast cancer, numerous studies have investigated an array of AhR ligands that promote antiproliferative, pro-apoptotic, antimigratory, or pro-differentiating effects [142]. Nonetheless, the role of AhR is not unambiguous. A recent and thorough review by Safe et al. summarized the numerous conflicting reports that exist on the role of AhR in breast cancer, as AhR has been associated with both tumor-suppressive and oncogenic functions [142], depending on the biological context, the applied ligand, the timing of the exposure, and the breast cancer model system (among other factors). AhR regulates the development of the mammary gland, as mice expressing low-affinity AhR^d^ alleles exhibit increased growth of alveoli relative to mice expressing the higher-affinity AhR^b^ allele [143].

In estrogen receptor α (ERα)-positive breast cancers, the ability of AhR to promote the degradation of ERα in a ligand-dependent manner drives antiproliferative effects, and this antiestrogenic activity represents one mechanism of tumor suppression [144]. Many triple-negative breast cancers (TNBCs) express high levels of AhR, and in patients with ER+ or ER-/PR- breast cancers, higher expression levels of AhR correlated with improved relapse-free survival relative to ER- and PR-negative cancers with a low expression of AhR [9]. With the goal of identifying AhR ligands that inhibit cancer growth and exhibit favorable toxicity profiles, we previously screened Food and Drug Administration-approved drugs and identified raloxifene as an AhR ligand with pro-apoptotic effects in triple-negative breast cancer (TNBC) cells [9], and further studies have identified a novel raloxifene analog with a more favorable toxicity profile than the parent molecule [145]. In addition to raloxifene and its analogs, we also identified the pre-clinical drug candidate CGS-15943 as an AhR-dependent pro-apoptotic molecule in TNBC [146], and recently discovered Analog 523 (a derivative of 11-Cl-BBQ) as a potent pro-apoptotic AhR ligand against TNBC cells and TNBC stem cells [147].

A recent study by Vogel et al. sought to determine the impact of the overexpression of AhRR (a target gene of AhR and a repressor of AhR’s transcriptional activity) on the progression of two breast cancer mouse models [148]: syngeneic E0771 breast cancer cell transplantation [149] (ERα-, PR+, HER2+), and the polyoma middle T antigen (PyMT) transgenic mouse model. Evidence supports the notion that AhRR acts as a tumor suppressor [150]. Vogel and colleagues found that the growth of E0771 was inhibited by the overexpression of AhRR, and TCDD-dependent tumor promotion was inhibited by the expression of AhRR as well. Similarly, the authors found that the overexpression of AhRR inhibited the growth of UCD-PyMT mammary tumor cells [148].

Future studies need to clarify the endogenous role of AhR in TNBCs, which are significantly heterogeneous in their molecular features and vulnerabilities [151,152]. The wide genetic and phenotypic variability in TNBC presents challenges for developing faithful models that recapitulate the progression of the disease. To examine the role of AhR even more rigorously in triple-negative breast cancer, future studies should determine the effect of deleting AhR, or the effect of AhR alleles with reduced activity [143] in a genetically engineered mouse model (GEMM) of triple-negative breast cancer [153,154]. Assessing differences in the tumorigenesis, global gene expression, protein expression, and metabolic alterations in these models would inform us of the role that AhR plays in the carcinogenesis of TNBC.

## 4. Crosstalk between Tumor Suppressor p53 and AhR in Cancer

To further define the role of AhR in cancer, we recently investigated the impact of the loss of AhR on tumorigenesis in p53-deficient mice [155]. *TP53*, encoding the tumor suppressor p53, is a transcription factor and critical regulator of the cell cycle, cell fate, cell death, and differentiation [156]. We reasoned that while the loss of AhR alone is insufficient to result in tumorigenesis, the ability of AhR to suppress carcinogenesis would become apparent in the background of disabled tumor suppressor signaling and the resulting oncogenic pressure downstream of the loss of p53.

Comparisons of AhR-expressing (*AHR*^+/+^) or AhR-deficient (*AHR*^−/−^ or *AHR*^−/+^) mice in p53 knockout or p53 heterozygous backgrounds revealed that the loss of AhR results in greater tumor numbers, a broader tumor spectrum, and significantly reduced survival relative to their AhR-expressing or AhR heterozygous counterparts in p53-deficient backgrounds (p53^−/−^ or p53^+/−^) (Figure 4). In p53 knockout backgrounds, AhR wild-type mice survived for 184 days while AhR knockout mice survived for only 127 days. The p53 knockout mice developed characteristic thymic lymphomas [155,157], and the AhR knockout mice in this background developed significantly more thymic lymphomas and leukemias [155]. Furthermore, AhR knockout mice featured an increased tumor spectrum, with growth of hemangiomas, adenocarcinomas, suspected cutaneous neoplasia, and suspected carcinomas, while AhR wild-type mice did not develop these [155]. Notably, the thymic lymphomas were strikingly larger in AhR knockout animals compared with their AhR wild-type and p53-deficient counterparts [155]. Kaplan–Meier analyses of the expression of AhR and p53 revealed that patients with high expression of AhR and p53 had improved survival outcomes in chronic lymphocytic leukemia, lung cancer, breast cancer, and metastatic melanoma.

The increased tumorigenesis in p53-deficient mice lacking AhR suggests that p53 and AhR may share common regulatory targets and that the expression of AhR is protective against tumorigenesis upon the loss of p53’s activity (Figure 5). In this respect, the relationship between AhR and p53 is epistatic, where the tumor-suppressive activity of AhR is observed upon mutation of the p53 gene. This phenomenon is supported by a recent study indicating that AhR is a key negative regulator of proliferation and tumorigenesis in p53 knockout cancer cells, which was identified by an unbiased genome-wide knockdown approach [158]. Both AhR and p53 can transcriptionally regulate the tumor suppressor p21^Cip1^ [28,79,94,159], pro-apoptotic factors such as Bcl-2 associated X-protein (Bax) [160,161], and insulin-like growth factor binding protein (IGFBP3) [162]. It is possible that AhR and p53 cooperate to co-regulate common tumor suppressor gene programs downstream of cellular stress events. Consistent with a functional crosstalk between p53 and AhR, we recently found that the antiproliferative activity of the AhR ligand 11-Cl-BBQ depended on the expression of p53 in addition to the induction of the expression of p27^Kip1^ [94].

## 5. AhR and Tumor Immunity

The aryl hydrocarbon receptor has key roles regulating the development and function of the immune system [95,99,163], and AhR has garnered significant attention in the fields of immunology and immuno-oncology for its role in immunosuppression downstream of its activation by tumor- and microenvironment-derived tryptophan metabolites such as kynurenine, quinolinic acid, and kynurenic acid, among others [164]. Kynurenine and related metabolites are generated as the catabolic products of tryptophan metabolism by indole-2,3,-dioxygenases 1 and 2 (IDO1), tryptophan dioxygenase (TDO2) [15], and, as more recently reported, IL4I1 [164,165,166]. Importantly, AhR is an upstream regulator of the expression of these metabolic enzymes. Broadly, kynurenine and the related metabolites activate AhR and can promote immunosuppression through an array of mechanisms, including the induction of CD4+ regulatory T cells [111], alteration of the function of dendritic cells [166], and numerous other outcomes (2). Significant effort has been invested into the blockade of this signaling axis, and multiple antagonists are being developed for their use alone and in combination with checkpoint blockade immunotherapies. Notably, certain clinical trials evaluating IDO1 inhibitors failed in terms of demonstrating therapeutic efficacy [167]. The recent identification of IL4I1 as a previously unappreciated tryptophan-catabolizing enzyme that produces kynurenine was proposed as a compelling explanation for this failure, and efforts are underway to target this enzyme [164,165].

Notably, there is evidence to support AhR as a positive regulator of immunity and the antitumor response in natural killer (NK) cells, dendritic cells, and other immune cell populations [168]. For example, the loss of AhR expression in NK cells resulted in reduced cytolytic activity and control of lymphoma growth, while in vivo administration of the endogenous ligand 6-formylindolo[3,2-b]carbazole (FICZ) enhanced AhR-dependent antitumor activity in these cells [168]. The pro-tumorigenic versus antitumor effects of AhR should also be considered in light of the cancer stage. For instance, as reviewed in Section 3, AhR’s activation and inhibition of inflammatory signaling in different immune cell populations opposed carcinogenesis in the intestine [124,125]. In contrast, with more advanced tumors, the effect of AhR in opposing the activation of CD8+ T cells via kynurenine signaling may be more dominant and associated with a negative response to immunotherapies. The response to AhR ligands is highly cell-specific. For instance, while kynurenine signaling has been dominantly characterized by its role in immune suppression, kynurenine was also found to have tumor-suppressive effects downstream of AhR in neuroblastoma, driving cellular differentiation [128]. Successful targeting of AhR will entail a consideration of the impact of AhR’s activation on immune signaling and will require the identification of biomarkers to predict the responses.

## 6. Tools and Therapies for Modulating the Function of AhR

While many FDA-approved drugs activate AhR, there is currently only one drug approved for targeting the function of AhR, with the relatively recent approval of tapinarof as an immunomodulatory AhR agonist for psoriasis [169,170,171]. For antagonizing the role of AhR in immune suppression, at least two different small-molecule AhR antagonists are currently in clinical trials: IK-175 and BAY2416964 [172,173]. Another AhR antagonist under clinical development is the molecule stemregenin-1 (SR-1) [81,174] for the purpose of expanding hematopoietic stem cells. There are currently no AhR agonists in clinical trials for cancer therapy; however, the activation of AhR by tryptophan is being explored clinically as a therapeutic approach for inflammatory bowel disease [175].

A large number of different AhR ligands have been explored by multiple laboratories for identifying molecules that exhibit favorable biological activity (e.g., tumor suppression and immunomodulation) and toxicity profiles through selective modulation of AhR-regulated transcription. We previously identified FDA-approved drugs such as raloxifene [9], leflunomide [140,141], flutamide [176], and two preclinical drug candidates, SU5416 [79] and CGS-15943 [146], as AhR ligands that drive antiproliferative or pro-apoptotic effects in different cancer cells via the receptor. Additionally, omeprazole and other AhR-active pharmaceuticals have been shown to exhibit anticancer activity [177,178]. To discover novel AhR ligands for therapeutic translation, our laboratory previously screened a small-molecule library and identified benzimidazoisoquinolines (BBQs) as a chemical class that possesses high affinity and is rapidly metabolized [46,94,110,112]. In vivo studies directed at immunomodulation demonstrated that these ligands were tolerated well [110,112]. Notably, we found that the benzimidazoisoquinoline 11-Cl-BBQ drove potent AhR-dependent antiproliferative effects in lung cancer cells [94]. Screening of 11-Cl-BBQ analogs led to the recent identification of Analog 523 as a potent AhR-dependent ligand with pro-apoptotic effects in TNBC cells and TNBC stem cells [147].

Another class of AhR ligands actively being researched include diet-derived, microbiome-derived, and endogenously derived indole compounds that are formed from the metabolism of tryptophan and tyrosine. While too extensive to completely review here, different tryptophan-derived ligands such as FICZ [179], 2-(10-H-indole-3-carbonyl) thiazole-4-carboxylic acid methyl ester (ITE) [180], and indole-3-carbinol (I3C) [181] have been explored as experimental and therapeutic AhR agonists [54] for cancer [5,10,182,183,184] and immune diseases [5]. The AhR ligand FICZ has been the subject of significant interest due to its high affinity for AhR, its endogenous origin, and the fact that it is efficiently and rapidly degraded by the CYP1 enzymes, enabling potent and transient activation of the receptor [179,184]. In addition to AhR knockout mice, AhR knockout zebrafish models have been generated [185,186,187]. Zebrafish studies interrogating the function of AhR have frequently used morpholino antisense oligonucleotides to knock down the expression of AhR [188,189]. One commonly used agent for antagonizing the activity of AhR is the chemical inhibitor CH223191 [190,191], which inhibits the activation of AhR by certain classes of AhR ligands (e.g., halogenated aromatic hydrocarbons), while it was unable to antagonize structurally distinct ligands such as flavonoids or indoles. Distinct from ligands that activate or inhibit the transcriptional activity of AhR, small-molecule chimeras or ‘molecular glues’ were recently developed to exploit the E3 ubiquitin ligase [144] function of AhR [192]. The study showed that a small-molecule chimera composed of the AhR ligand ITE linked to retinoic acid could promote the degradation of cellular retinoic acid-binding protein (CRABP) via the E3 ligase activity of AhR [193], and additional AhR-dependent chimeras were also successfully used.

## 7. Discussion

Conflicting reports on the functions of AhR in cancer emphasized the necessity of understanding the mechanisms and the biological contexts (tissues and genetic alterations) that drive AhR-dependent tumor suppression. This review sought to summarize and discuss the various lines of evidence supporting AhR-dependent tumor-suppressive effects, with an emphasis on studies where endogenous AhR signaling opposed carcinogenesis in vivo (Figure 3).

Through various molecular mechanisms, the activation or expression of AhR inhibits carcinogenesis (Figure 6). Clearly, AhR-dependent control of the Wnt/β-catenin signaling pathway represents an important mechanism through which the receptor promotes tumor suppression in at least liver and intestinal cancers. This is supported by Wnt/β-catenin-driven cancer models with mutations of the tumor suppressor APC and the deletion of AhR [83,84], and additional mouse cancer models where the loss of AhR activity resulted in enhanced tumorigenesis and elevated Wnt signaling [115,116,117,125,131]. In the colon, AhR has dual roles in tumor suppression, by promoting the integrity of the epithelial barrier, dampening inflammation, and antagonizing proliferation signals downstream of Wnt/β-catenin during the regenerative process. In multiple cancer models, AhR restricts stem cell proliferation by repressing pluripotency factors such as Oct-4, Sox2, c-Myc, and Nanog [88,105,131], and the expression or activation of AhR can promote differentiation in multiple cancer types [86,87,90,122,125,128,129]. Additionally, AhR can oppose oncogenic pathways including PI3K-AKT growth factor signaling [104,182], SHH, and TGF-β signaling [86].

The loss of AhR expression significantly increases tumorigenesis in p53-deficient backgrounds [155], supporting functional crosstalk between these two transcription factors. The tumor-suppressive role of AhR is underscored by the fact that a single allelic copy of AhR is protective against tumorigenesis [155], and the complete loss of AhR’s expression drives an increased spectrum of tumors not observed in AhR-expressing, p53-deficient animals. Transcription factors such as p53, ATF3/ATF4, AhR, nuclear factor erythroid 2–related factor 2 (Nrf2), and heat-shock factor 1 (HSF1) all function as mediators of the cellular stress response and share common regulatory targets [192]. In lung cancer cells, Nothdurft et al. found that the AhR-dependent induction of ATF4 drove tumor suppression [106]. We found that the endogenous tumor-suppressive effects of AhR became prominent in the background of p53 deficiency, suggesting that AhR partially compensates for the loss of p53 activity [155]. Both AhR and p53 share common regulatory targets such as p21^Cip1^, IGFBP3, and Bax, and it is possible that AhR maintains the expression of certain gene targets of wild-type p53 in the absence of p53. Restoring the expression of p53-regulated tumor suppressor genes in p53 mutant or deficient cancers through the induction of stress response factors such as ATF3 and ATF4 represents an area of active investigation [194,195], and it is possible that AhR cooperates with these factors to inhibit carcinogenesis.

Delineating the molecular contexts where AhR exerts tumor suppression is important for both chemoprevention and therapeutic targeting of the receptor. Understanding the molecular determinants of tumor suppression in response to the activation of AhR is particularly salient when considered against the background of kynurenine–AhR-dependent immune suppression. In this respect, identifying the cancer types that respond positively to the activation of AhR, identifying overexpressed or silenced factors that modulate the response of AhR, and identifying the stage(s) of tumorigenesis when the activation of AhR drives the inhibition of growth is paramount. The fact that AhR intersects with multiple oncogenic signaling pathways suggests that the activation of AhR may offer therapeutic vulnerability in cancer cells where individual pathway components are not easily targeted or ‘druggable’ (e.g., c-MYC). Further, unrealized therapeutic opportunities may exist to exploit the activation of AhR in cancer types that feature the elevated activity of pathways controlled by the receptor. For instance, cancers that feature elevated Wnt signaling or acquire resistance to existing Wnt pathway inhibitors could be candidates for AhR-targeted therapeutics.

## 8. Conclusions

In vivo evidence has demonstrated that AhR can function as a tumor suppressor in multiple cancers, including lung, breast, liver, prostate, skin, and different cancers of the intestine, hematopoietic system, and brain (Figure 3). The expression of AhR restricts carcinogenesis in different tissues through the suppression of oncogene activity and the induction of growth inhibitory gene programs (checkpoint activation, cell death) and cooperativity with other tumor suppressors such as p53. AhR has both p53-dependent and p53-independent antiproliferative functions. By highlighting the targets and biological contexts of AhR-dependent tumor suppression, this review also provides a framework for combination studies and investigations of novel therapeutic approaches via the modulation of AhR.

AhR represents a therapeutically viable cancer target, as select AhR ligands can inhibit the proliferation and migration of cancer cells, induce cellular differentiation, and promote apoptosis. The identification of AhR ligands that selectively inhibit the growth of cancer cell growth while exhibiting favorable toxicity profiles has been the subject of extensive research in our laboratory [9,46,53,54,69,79,94,112,140,141,145,146,155,176] and others [196,197]. Toward this, further mechanistic details regarding the molecular targets downstream from AhR will enable greater predictive power required for therapeutic translation.

## Figures and Tables

**Figure 1 biology-12-00526-f001:**
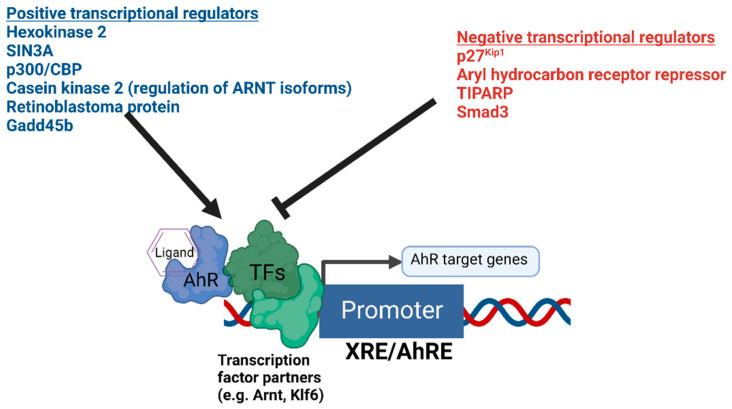
Summary of the positive and negative transcriptional co-regulators of AhR signaling. The diagram depicts AhR bound to DNA in complex with potential heterodimer partner proteins and regulators. Different positive and negative regulators of AhR-mediated transcription are summarized above.

**Figure 2 biology-12-00526-f002:**
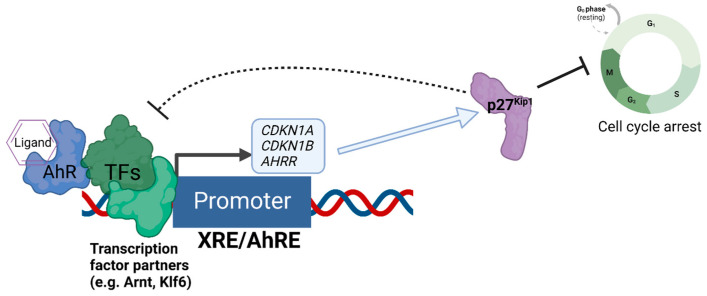
Negative regulation of AhR-mediated transcription by the expression of p27^Kip1^. The diagram depicts the regulatory relationship between p27^Kip1^ and AhR, whereby AhR acts as a direct transcriptional regulator of the transcription of p27^Kip1^, and increased expression of p27^Kip1^ both inhibits cell cycle progression and represses the AhR-mediated transcription of genes such as *CDKN1A* (encoding p21Cip1), *AHRR*, and *CDKN1B* (encoding p27^Kip^1 itself) as an auto-feedback loop.

**Figure 3 biology-12-00526-f003:**
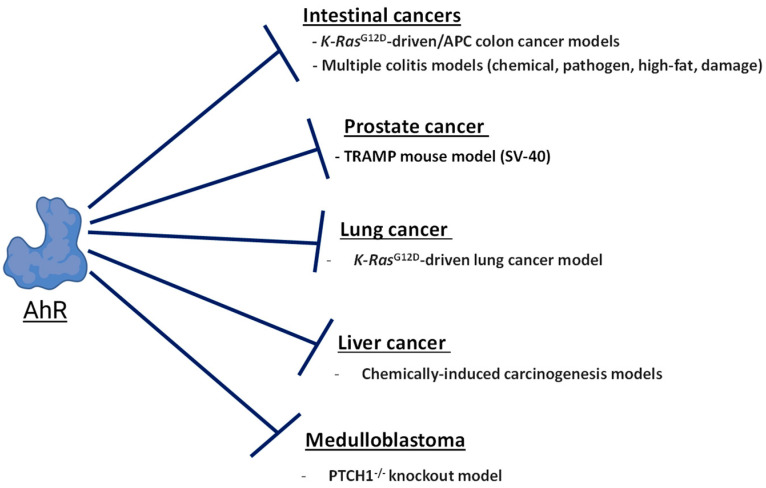
The role of AhR in tumor suppression. The diagram shows a summary of mouse cancer models where the expression of AhR opposes carcinogenesis.

**Figure 4 biology-12-00526-f004:**
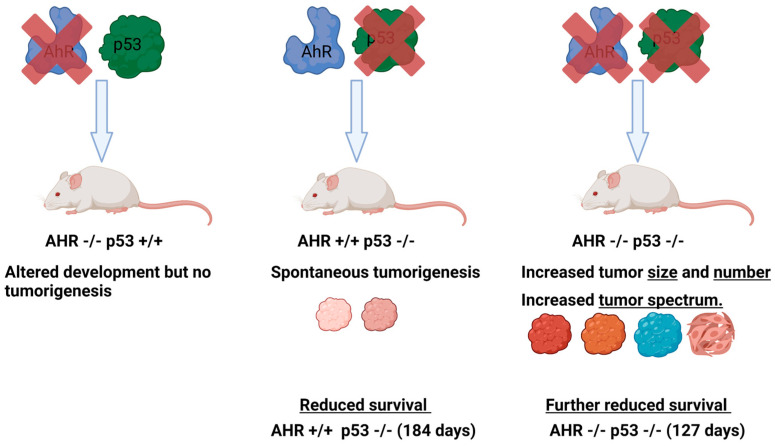
Loss of AhR leads to increased tumorigenesis in p53deficient mice. Diagram representing the relative effects of AhR deletion, p53 deletion, or the deletion of both genes on tumorigenesis, tumor spectrum, and survival. Diagrams and annotations refer directly to experimental data published in [155].

**Figure 5 biology-12-00526-f005:**
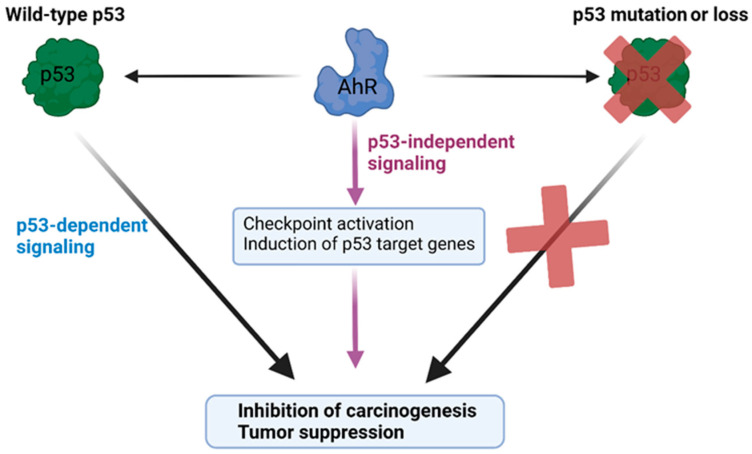
Model of the cooperative roles of AhR and p53 in tumor suppression. The diagram depicts a model of p53 and AhR cooperating in restraining carcinogenesis. AhR exerts both p53-dependent and p53-independent tumor-suppressive activities. The activation of AhR can promote p53-dependent antiproliferative effects, and, in the absence of wild-type p53 activity, the expression of AhR is protective against carcinogenesis and exerts tumor suppression via p53-independent gene programs.

**Figure 6 biology-12-00526-f006:**
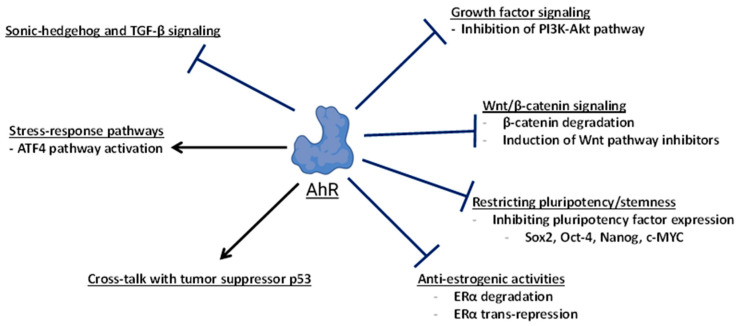
Pathways and targets of AhR-dependent tumor suppression. This diagram broadly summarizes the mechanisms and molecular targets of AhR involved in suppressing carcinogenesis or tumor growth.

## Data Availability

Not applicable.

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
