# Peer review of "Tumor-Suppressive Functions of the Aryl Hydrocarbon Receptor (AhR) and AhR as a Therapeutic Target in Cancer"

_biology, 2023, doi:10.3390/biology12040526_

Round 1
Reviewer 1 Report
This is a well-written and useful review of the role of AHR in the development of a variety of cancers. It will be useful to many investigators in the AHR field in particular. My only comment/correction is that on line 74 p450 should be P450.
Author Response
Reviewer 1: We would like to thank the Reviewer for dedicating their time and and providing helpful feedback.
Comment 1: This is a well-written and useful review of the role of AHR in the development of a variety of cancers. It will be useful to many investigators in the AHR field in particular. My only comment/correction is that on line 74 p450 should be P450.
Response: Thank you for this correction. It has been changed.
Reviewer 2 Report
Elson et al., presented a comprehensive review of the potential role of the aryl hydrocarbon receptor (AhR) in the tumorigenesis of a variety of cancer types. The manuscript is well-written and provides an update on AhR and contributes significantly to the literature. A few things to be addressed before considering the manuscript for publication.
1- Authors mentioned several times the therapeutic targeting of AhR. However, I did not see any discussion of these potentials or modern technologies that can be utilized to utilize AhR suppression and how this can be tested at the level of animal experiments and/or clinical trials (for example, siRNA, antibodies and gene editing)
2- "While bearing growth defects, 205 AhR-null mice do not spontaneously develop tumors. This indicates that the endogenous tumor suppressive activities of AhR become apparent in the context of oncogene activation and disabled tumor suppressor gene function" This sentence is unclear whether AhR is generally tumor suppressive or oncogene
3- section 2.1 Role of AhR in xenobiotic metabolism. I suggest this section to be removed since it is irrelevant to AhR and cancer progression.
4- please provide a figure showing how AhR signaling, along with its co-regulator, regulate several pathways involved in tumor development
5- 3.9. Crosstalk between tumor suppressor p53 and AhR in cancer. This should be a separate section along with other general mechanistic contributions to tumor suppression.
6- few spelling and grammatical errors
Author Response
Reviewer 2: We would like to thank the Reviewer for dedicating their time in reviewing our manuscript, providing helpful feedback to improve various sections.
Elson et al., presented a comprehensive review of the potential role of the aryl hydrocarbon receptor (AhR) in the tumorigenesis of a variety of cancer types. The manuscript is well-written and provides an update on AhR and contributes significantly to the literature. A few things to be addressed before considering the manuscript for publication.
- Authors mentioned several times the therapeutic targeting of AhR. However, I did not see any discussion of these potentials or modern technologies that can be utilized to utilize AhR suppression and how this can be tested at the level of animal experiments and/or clinical trials (for example, siRNA, antibodies and gene editing)
Thank you for this suggestion. While the scope of this section was not comprehensive, given the breadth of available studies, we have included a section focused on tools and therapies for modulating AhR activity.
- "While bearing growth defects, 205 AhR-null mice do not spontaneously develop tumors. This indicates that the endogenous tumor suppressive activities of AhR become apparent in the context of oncogene activation and disabled tumor suppressor gene function" This sentence is unclear whether AhR is generally tumor suppressive or oncogene.
This sentence was edited to be clearer – changed growth defects to ‘altered development’.
- section 2.1 Role of AhR in xenobiotic metabolism. I suggest this section to be removed since it is irrelevant to AhR and cancer progression.
We removed the majority of this section but retained enough background about xenobiotic metabolism and CYP1 enzymes so that later sections (e.g. touching on regulation of CYPs as measure of AhR activity) are easily understood
- please provide a figure showing how AhR signaling, along with its co-regulator, regulate several pathways involved in tumor development
To address this comment, we decided to generate two different figures that summarize the impact of co-regulators on AhR signaling. Figure 2 details positive and negative regulators of AhR transcriptional activity. Figure 3 depicts the regulation of AhR activity by p27 specifically. The exact downstream pathways regulated by different co-activators/repressors of AhR are not established well-enough to confidently depict them in these figures.
- 9. Crosstalk between tumor suppressor p53 and AhR in cancer. This should be a separate section along with other general mechanistic contributions to tumor suppression.
To better organize this section within the review, the subsection of p53 and AhR crosstalk was converted to its own Section, as it did not ‘fit’ into the upstream organization of the review by cancer type.
- few spelling and grammatical errors
We have gone through and reviewed for errors.
Reviewer 3 Report
The manuscript " A Versatile Receptor with a Penchant for Surprise: The Aryl hydrocarbon receptor (AhR) as a Therapeutic Target in Cancer" reviewed the roles of AhR in the pathogenesis, development and treatment of cancers. The paper was organized properly. However, there are some minor comments.
1. Authors only reviewed the roles of AhR in the cell proliferation, apoptosis, migration of cancer cells. AhR is also involved in the tumor immunity. The roles of AhR in tumor immunity should be included.
2. Whether there are drugs targeting AhR for the treatment of cancer. Authors should concluded the drugs in clinical phase for cancer treatment by targeting AhR.
Author Response
Reviewer 3: We would like to sincerely thank Reviewer 3 for their dedicated time in reviewing our manuscript. We have worked to address comments and improve the review.
The manuscript " A Versatile Receptor with a Penchant for Surprise: The Aryl hydrocarbon receptor (AhR) as a Therapeutic Target in Cancer" reviewed the roles of AhR in the pathogenesis, development and treatment of cancers. The paper was organized properly. However, there are some minor comments.
- Authors only reviewed the roles of AhR in the cell proliferation, apoptosis, migration of cancer cells. AhR is also involved in the tumor immunity. The roles of AhR in tumor immunity should be included.
We have included a section on AhR and tumor immunity, seeking to succinctly summarize both the pro- and anti-tumor effects of AhR in immune signaling. Thank you for this helpful suggestion.
- Whether there are drugs targeting AhR for the treatment of cancer. Authors should concluded the drugs in clinical phase for cancer treatment by targeting AhR.
We appreciate this helpful comment. We have included a corresponding section on AhR-targeted drugs and those in clinical trials. Reviewer 2 suggested a section detailing tools for modulating AhR function, so we have combined Therapies and Tools targeting AhR into one Section.
Round 2
Reviewer 2 Report
The authors have addressed my comments and I believe the manuscript can be considered for publication